# A Straightforward Cytometry-Based Protocol for the Comprehensive Analysis of the Inflammatory Valve Infiltrate in Aortic Stenosis

**DOI:** 10.3390/ijms24032194

**Published:** 2023-01-22

**Authors:** Pablo Álvarez-Heredia, José Joaquín Domínguez-del-Castillo, Irene Reina-Alfonso, Carmen Gutiérrez-González, Fakhri Hassouneh, Alexander Batista-Duharte, Antonio Trujillo-Aguilera, Rosalía López-Romero, Ignacio Muñoz, Rafael Solana, Alejandra Pera

**Affiliations:** 1Immunology and Allergy Group (GC01), Maimonides Biomedical Research Institute of Cordoba (IMIBIC), University of Cordoba, Reina Sofia University Hospital, Avda Menedez Pidal s/n, 14004 Cordoba, Spain; 2Cardiovascular Pathology Group (GA09), Maimonides Biomedical Research Institute, Reina Sofia University Hospital, University of Cordoba, Avda Menedez Pidal s/n, 14004 Cordoba, Spain; 3Immunology and Allergy Service, Reina Sofia University Hospital of Cordoba, Avda Menedez Pidal s/n, 14004 Cordoba, Spain; 4Department of Cell Biology, Physiology and Immunology, University of Cordoba, Avda Menedez Pidal s/n, 14004 Cordoba, Spain

**Keywords:** flow cytometry, protocol, aortic stenosis, inflammation

## Abstract

Aortic stenosis (AS) is a frequent cardiac disease in old individuals, characterized by valvular calcification, fibrosis, and inflammation. Recent studies suggest that AS is an active inflammatory atherosclerotic-like process. Particularly, it has been suggested that several immune cell types, present in the valve infiltrate, contribute to its degeneration and to the progression toward stenosis. Furthermore, the infiltrating T cell subpopulations mainly consist of oligoclonal expansions, probably specific for persistent antigens. Thus, the characterization of the cells implicated in the aortic valve calcification and the analysis of the antigens to which those cells respond to is of utmost importance to develop new therapies alternative to the replacement of the valve itself. However, calcified aortic valves have been only studied so far by histological and immunohistochemical methods, unable to render an in-depth phenotypical and functional cell profiling. Here we present, for the first time, a simple and efficient cytometry-based protocol that allows the identification and quantification of infiltrating inflammatory leukocytes in aortic valve explants. Our cytometry protocol saves time and facilitates the simultaneous analysis of numerous surface and intracellular cell markers and may well be also applied to the study of other cardiac diseases with an inflammatory component.

## 1. Introduction

Aortic stenosis (AS) is one of the most common and serious valve pathologies in developed countries. AS is characterized by the thickening and calcification of the aortic valve leaflets. It can be secondary to rheumatic inflammation or congenital. Age-degenerative calcific aortic stenosis (also known as senile AS or sclerocalcific) is currently the most frequent cause of adult aortic stenosis in the USA and Western Europe. Up to 30% of adults over 65 years have valvular sclerosis, and the incidence of all-cause death from cardiovascular disease due to aortic valve sclerosis has increased by 35% [1]. With time, most aortic sclerosis patients progress to obstructive aortic stenosis. In the United States, calcified aortic valve stenosis has become the major indication for aortic valve replacement, and it is the third-most common cardiovascular disease after coronary heart disease and hypertension [2]. In Europe, 82% of aortic stenosis is caused by calcification [3]. Most AS patients (80%) progress to cardiac insufficiency, valve replacement, or death within a period of 5 years [4,5].

Diagnostic suspicion of AS is mainly clinical (electrocardiography and echocardiography). Generally, it does not lead to hemodynamic compromise in the adult until the valvular hole narrows below ø 1 cm^2^, leaving the patient asymptomatic despite having critical AS [6]. Most AS patients suffer a gradual progressive obstruction over the years and symptoms are understated. Thus, there is an evident necessity for biomarkers for the prompt detection of this pathology in earlier stages of the disease.

Although aortic valves experience mechanical stress throughout life, with aging the triggering factor, not all older individuals develop AS [7,8]. Therefore, other factors must play a key role in this progression. AS is associated with a higher risk of coronary arteriopathy events. Histopathologic analysis shows alterations in the valves similar to those observed in atherosclerosis and vascular inflammation. Over the past years, results obtained from different studies indicate that AS could be related to coronary atherosclerosis, with both diseases triggered by similar pathogenic mechanisms [9,10,11,12]. Thus, it is probable that AS is an atherosclerotic-like process, not only in tricuspid patients but also in bicuspid ones [13,14,15]. Furthermore, there is an association between AS and atherosclerosis risk factors—age, sex, smoking, diabetes mellitus, hypertension, elevated low-density lipoproteins, (LDL), reduced high-density lipoproteins (HDL), and increased C-reactive protein (CRP) [16]. Additionally, it has been demonstrated that there is a strong association between AS severity and the presence and severity of aortic atheromas, which indicates that AS could be a manifestation of the atherosclerotic process.

Immunohistochemical analysis has shown the presence of B cells, T cells (CD4+ y CD8+), and other immune cells (macrophages, mastocytes) in the inflammatory infiltrate of the valve [17,18,19,20]. It is probable that with age, mechanical stress alters the integrity of the valvular endothelium leading to lipoprotein accumulation that induces the expression of adhesion molecules (ICAM-1, VCAM-1) in the endothelium, facilitating T cell and macrophage infiltration through blood and lymphatic vessels [19,21]. Furthermore, it has been shown that M1 macrophages promote calcification of the aortic valve [22]. Inflammatory cells release mediators that stimulate the destruction of the normal collagen field and elastic fibers in the aortic valve. Later, fibroblasts differentiate into myofibroblasts and these into osteoblast-like cells, leading to the calcification of the valve. Normal aortic valves do not have microvessels. However, during the calcification process, new microvessels are formed, which attract more pro-inflammatory cells (for revision, see [1] and [23,24]). Furthermore, molecular analysis has shown that most T cell populations present in the valvular infiltrate of AS patients consist of oligoclonal expansions, and some of the infiltrating clones correspond with the ones found in peripheral blood, corroborating the existence of cellular traffic between the periphery and the valve [20,25].

These results indicate that aortic valve stenosis is an active inflammatory process. However, up to date, there are no effective treatments except the surgical or transcatheter replacement of the valve itself. For that reason, a thorough characterization of the cell populations present in the valvular infiltrate and the peripheral blood of AS patients is of utmost importance for the development of new effective treatments to avoid surgical intervention in these patients, as well as for the discovery of biomarkers for early diagnosis. Here we present a novel powerful cytometry-based protocol, which allows both the phenotypical and functional characterization of the immune cells present in the aortic valve inflammatory infiltrate.

## 2. Results

A flow cytometric technique was used to characterize circulating and infiltrating immune cells of AS patients. Histological and immunohistochemical (ICH) techniques were used as controls for comparison with the flow-cytometry based protocol.

### 2.1. Characterization of Infiltrating and Circulating Immune Cells by Flow-Cytometry

Aortic stenosis is characterized by the thickening and accumulation of calcium deposits in the aortic valve leaflets. After digestion of the valvular tissue, samples go through to rounds of filtering to clean the calcium debris (70 and 20 μm ∅ cell restrainers). However, a large amount of calcium debris will remain in it. During sample acquisition in the BD LSR Fortessa SORP cytometer, this debris hinders the immune cell analysis. To solve this visualization problem, we used a LIVE/DEAD™ Fixable Far Red Dead cell stain (655 nm emission maximum) that, when confronted with a stain for anti-CD45 in AF700 (719 nm emission maximum), allowed us to properly discriminate the live leukocytes present in the sample from the debris, facilitating the identification of our target cells. After the live-leukocyte gate, anti-CD8 in V500 (500 nm emission maximum) was confronted with anti-CD45 in AF700 to remove any remaining contamination, which exhibited high autofluorescence in the V500 channel. Finally, singlets were gated in an FSC-H vs. FSC-A plot. After the data-cleaning process, infiltrating leukocyte subpopulations were identified and the phenotypic characterization of T cells was performed using the gating strategy shown in Figure 1. The gating strategy for the immunophenotyping of peripheral cell subpopulations was carried out as depicted in Appendix A.

### 2.2. Frequency of Circulating and Infiltrating Leukocyte Subpopulations

The cytometry panel for aortic-valve-infiltrating cells included phenotypic markers for the identification of monocytes, neutrophils, NK cells, and T cells (Appendix A). The frequencies of infiltrating leukocyte subpopulations were compared to those obtained from the peripheral blood of AS patients (Figure 2 and Figure 3, Appendix A).

The results showed that, compared with peripheral blood, within the valvular infiltrate there was a higher frequency of monocytes, total NK cells, CD56^bright^ NK cell subset, total T cells, CD8 T cells, and CD4+CD8+ T cells. Conversely, there was a lower frequency of CD16+ neutrophils, CD4+ T cells, and CD56^dim^ NK cells (Figure 2). Moreover, infiltrating CD4+ T cells have a higher expression of CD56 and a lower expression of CD28 than in the periphery (more differentiated) (Figure 3A). In the CD8+ T cell subset, there was a higher proportion of CD28^null^ cells in the infiltrate versus the periphery but with a lower expression of CD56 (Figure 3B). Finally, the phenotypic analysis of CD4−CD8− T cells, mostly TCRγδ cells, showed that the proportion of CD56-expressing cells was higher in the periphery than in the valve (Figure 3C).

### 2.3. Functional Analysis

We further validated the feasibility of performing ICS and functional analysis of valve-infiltrating cells. For the functional analysis we studied the response (production of IFN-γ, TNF-α or granzyme B) of the infiltrating T cells to a polyclonal stimulus (Cytostim). As the number of infiltrating T cells is limiting, PBMCs from the same patient were used as a negative control. After incubation cells were analyzed by flow cytometry (Appendix A). Our data show that our protocol can be used to study intracellular markers such as granzyme B (Figure 4). Moreover, our results demonstrate that despite the low number of cells, our protocol is sensitive enough to detect cytokine production by infiltrating cells.

## 3. Discussion

Numerous previous studies have attempted to characterize the leukocyte populations of the valvular infiltrate [18,20,26]. However, the limitations of IHC technique prevent a thorough and accurate characterization. Immunohistochemistry requires serial histological sections to determine each of the valve-infiltrating leukocyte subpopulations. The number of staining antibodies to be used per slice and the characterization of co-expression markers is also limited.

Our novel protocol changes the paradigm in the characterization of the aortic valve inflammatory infiltrate. The methodology for the isolation of the valvular infiltrating cells and the flow-cytometry-based techniques applied allowed us to identify and quantify several valvular-infiltrating populations simultaneously, providing further information regarding their phenotype. All experiments were performed using fresh samples. Thus, future studies are needed to confirm if this protocol has the same efficiency in frozen samples. Nevertheless, the application of this protocol reduces sample processing times and complexity, allows the performance of functional assays, and the study of intracellular markers. Thus, our protocol opens the possibility of studying not only which cell subsets are implicated in the AS pathophysiology process, but also gives information about what these cells can do and in response to which stimuli. Moreover, our technique gives information of the total aortic valve infiltrate and not only of a section as in the immunohistochemistry methods.

Our study provides a more objective information regarding the immunopathology of the aortic stenosis than previous IHC-based studies. Our results showed a predominancy of T cells and monocytes in the valve infiltrate compared to peripheral blood, confirming the relevance of the inflammatory process in the development of the disease (Figure 3 and Appendix A). In this sense, it has been suggested that infiltrating innate immune cells such as monocytes/macrophages or neutrophils trigger the process of valve degeneration (calcification), while adaptive immune cells will contribute to chronic inflammation, aggravating the pathology [24]. Our data confirm the prevalent presence of monocytes/macrophages and T cells in the valve infiltrate and opens the possibility of performing future full phenotypic and functional characterization of these cell subpopulations. Besides, our results corroborate previous molecular data on the CD4:CD8 T cell ratio in the valve infiltrate [18], but show that there is a greater frequency of CD4 T cells here than in the periphery, indicating that these cells could have a more important role than originally thought. Furthermore, our data confirmed that in the valve infiltrate there is a predominance of CD28^null^ CD8+ T cells as shown in molecular studies [20] and we extend these results, showing that this is also true for the CD4+ T cell subset.

All this evidence shows that our protocol offers a technical advantage over other methodologies for the study of cardiovascular, and other tissues’, immunopathology. An in-depth characterization of the cell populations implicated in AS will allow the identification of new biomarkers of early diagnosis and the development of new therapies for the treatment of this disease.

## 4. Materials and Methods

### 4.1. Subjects

A total of 50 calcified human aortic tricuspid valves were included in the study (Table 1). Calcified aortic valves and peripheral blood samples were harvested, by the Cardiovascular Surgery Unit of RSUH, from aortic stenosis (AS) patients undergoing aortic valve replacement, at the time of cardiovascular surgery. AS patients’ selection was done according to the following criteria: patients with aortic stenosis (isolated or in association with others cardiac diseases) with indication for surgical replacement, regardless of the anatomical characteristics of the native valve. Exclusion criteria included: chronic infections apart from CMV (HCV, HBV, HIV, etc.), inflammatory disease (oncologic diseases, rheumatoid arthritis, ankylosing spondylitis, ulcerative colitis, Crohn’s disease, celiac disease, systemic lupus erythematosus, multiple sclerosis, psoriasis, etc.), immunosuppressor treatment or hemodialysis. antihypertensive drugs that have calcium channel blockers as their active ingredient, such as Lercanidipine, Felodipine, Amlodipine, Bepridil, Diltiazem, Isradipine, Nicardipine, Nifedipine, Nimodipine, Verapamil, Clevidipine and Nisoldipine (only for those included in the functional analyses).

All donors were informed, and signed informed consent was obtained to participate in the study. The study was approved by the Ethics Committee of Hospital Universitario Reina Sofia of Cordoba (Spain).

For the surgical valve replacement procedure, during anesthetic induction, the main drug used was Etomidate (20 mg). To ensure correct anesthetic levels during the whole procedure the surgical team used Propofol at low doses and F (300 µg) in continuous infusion at high doses. Other drugs used during induction were Midazolam (1 mg/mL), Cisatracurium (20 mg), or Remifentanil (15 mg). In addition, analgesic drugs (Lidocaine) and neuromuscular blockers (Rocuronium) were also used. During the surgical procedure and extracorporeal pump time, several drugs were used to ensure the correct hemodynamic stability. The most common were: antiarrhythmics (Bisoprolol or Amiodarone), inotropes (Dobutamine, Levosimendan or Milrinone), chronotropes (adrenaline), vasodilators (nitroglycerin), and vasopressors (noradrenaline).

### 4.2. Reagents

RPMI 1640 medium (PanBiotech, cat. P04-17500. Aidenbach, Germany);Phosphate-buffered saline (PanBiotech, cat. P04-36500. Aidenbach, Germany);Fetal Bovine Serum (Gibco, ref. 10270106. Waltham, USA, Massachusetts);Collagenase D 0.24U/mg lyophilized (Roche Diagnostics, ref. 11088866001. Basel, Switzerland);FcBlock (BD Becton Dickinson, cat. 564220. Franklin Lakes, USA, New Jersey);MACSQuant Running Buffer (Miltenyi, cat. 130-092-747. Bergisch Gladbach, Germany);Brefeldin A/Golgi Plug (BD Becton Dickinson, cat. 555029. Franklin Lakes, USA, New Jersey);Cytostim human (Miltenyi, cat. 130-092-173. Bergisch Gladbach, Germany);EDTA (PanReac AppliChem, cat. A4892. Chicago, USA, Illinois);Glutamine (Biowest, cat. X0550-100. Nuaillé, France);Penicillin/Streptomycin (Biowest, cat. L0022-100. Nuaillé, France).

The following materials were used to process the sample:15 mL tubes with strew cap (NerbePlus, ref. 02-502-8001. Winsen, Germany);50 mL tubes with strew cap (NerbePlus, ref. 02-572-8001. Winsen, Germany);5 mL Flow Cytometry tubes (Corning, ref. 352052. Corning, USA, New York);70 μm tube filter (Biologix, ref. 15-1070. Jinan, Shandong, China);20 μm tube filter (Pluriselect, ref. 43-10020. Leipzig, Alemania);Scalpel Blades (Heinz Herenz, ref. 1110923. Hamburg, Germany);Needle (BD Becton Dickinson, ref. 305899. Franklin Lakes, USA, New Jersey);Syringe (BD Becton Dickinson, ref. 300928. Franklin Lakes, USA, New Jersey);Sterile Petri dish (Deltalab, ref. 200219. Barcelona, Spain);Sterile 5 mL Flow Cytometry tubes (Corning, ref. 352054. Corning, USA, New York,).

### 4.3. Reagent Preparation

For the cell isolation protocol, 6 mL of 2% Collagenase D in RPMI-1640 was prepared in a 15 mL tube (digestion solution). Collagenase was reconstituted following the manufacturer’s instructions. For functionality assay, 1 mL of 10% FBS, 1% glutamine, and 1% penicillin/streptomycin in RPMI-1640 (culture medium) was prepared in a 15 mL tube.

### 4.4. Surgical Procedure for Aortic Valve Tissue Collection

For aortic valve replacement surgery, it is necessary to open the chest and see the heart. This can be done through sternotomy, mini-sternotomy, or anterior thoracotomy. Once the skin, the subcutaneous cellular tissue, and the corresponding bone access have been opened, a retractor is placed to allow correct exposure throughout the surgery. Next, the pericardium is sectioned perpendicularly and pulled towards the retractor, elevating the cardiac structures. To be able to work on the aortic valve it is necessary to do it under cardiac ischemia and using an extracorporeal circulation (ECC) pump. The ECC pump requires certain levels of anticoagulation to be maintained during the procedure; we use heparin. Once heparin is administered the cannulas are inserted. From this moment on, all the patient’s cardiac output is handled by the ECC pump. To stop the heart, an aortic clamp is placed between the arterial perfusion cannula and the cardioplegia cannula. Once the heart has stopped, an aortotomy is performed about 2–3 cm above the valve annulus. At this time, the process of extracting the valve tissue begins, starting with the non-coronary leaflet, which is the closest. This methodology will limit the manipulation of the aortic leaflets to the minimum, since sometimes calcium deposits are fragile and easily disintegrate. Resection is performed by gentle traction in the central area of the leaflet, using a scalpel or dissection scissors. The cut will start at the commissure closest to the surgeon and extend along the lower edge of the leaflet located next to the aortic annulus. This technique ensures the obtention of the maximum amount of tissue. The procedure is repeated for each of the three leaflets. After extraction, leaflets are placed in a sterile tray with a physiological saline solution. Finally, the collected tissues are moved into RPMI solution for transportation and study in the laboratory.

### 4.5. Histological and Immunohistochemical (IHC) Analysis

One aortic valve from an AS patient was processed with this methodology. After surgical sample collection, valve leaflets were fixed in 10% buffered formalin for 24 h and processed for paraffin embedding using standard procedures for histological and IHC analysis. The specimens were sectioned at 4 µm thickness and stained with haematoxylin–eosin (H&E). Decalcification with a 10% EDTA solution was done if needed.

For IHC analysis, 3 µm-thick sections were obtained, and immunostaining was performed using the rabbit monoclonal antibody anti-CD3 (1:150) (Abcam ab16669, Cambridge, UK) with the envision FLEX/HRP system (Dako, Glostrup, Denmark). For IHC staining, the secondary antibody (Envision FLEX/HRP) was used for 30 min at room temperature, followed by 3,3′-diaminobenzidine (DAB) staining (Dako, Glostrup, Denmark) before being counterstained with Harris haematoxylin. Human lymph node was used as a positive control for CD3 antibody and staining in the absence of the primary antibody was used as a negative control. IHC images were acquired using a Leica microscope (Leica Microsystems, Wetzlar, Germany) and a Thunder Imager microscope (Leica Microsystems, Wetzlar, Germany), respectively.

### 4.6. Cell Isolation Protocol

To prepare a single-cell suspension for flow cytometry, aortic valve leaflets were processed using the following protocol:Wash leaflets 3–5 times with 10 mL of PBS + 1% FBS, to ensure no peripheral blood contamination.Inject 1 mL of digestive solution per 1 cm of valve leaflet and incubate for 5 min at room temperature.Cut the valve tissue using the scalpels to avoid calcification.Transfer the processed tissue to the digestion solution tube and incubate it at 37 °C for 1 h and 30 min under continuous stirring.Filter the digested tissue through a 70 μm ∅ cell restrainer to a 50 mL tube and top up with 10 mL of washing solution.Centrifuge the cells for 5 min at 400× *g*, then discard the supernatant and vortex gently.Add 4.5 mL of PBS + 1% FBS and filter the cells through a 20 μm ∅ cell restrainer to a sterile 5 mL FACS tube.Centrifuge the cells for 5 min at 400× *g*, then discard the supernatant and vortex gently.Continue with surface or intracellular staining protocol.

### 4.7. Surface Cell Staining for Flow Cytometry

Before staining, 0.5 μL of Fc Block was added and incubated at room temperature for 10 min to reduce non-specific antibody staining. Isolated cells were directly stained with the cocktail of antibodies corresponding to the valve panel (Appendix A) or the functional assay panel (Appendix A) and incubated for 20 min at RT in the dark. Finally, cells were resuspended in 500 μL of Running Buffer. Of note, no wash step was performed after staining to obtain a maximum number of cells.

### 4.8. Functionality Assay

For functionality assays, isolated infiltrating cells and peripheral blood mononuclear cells (PBMCs) were processed using the following protocol:Add 250 μL of culture medium to the cell suspension in a sterile FACS tube. For PBMCs functionality assay, 2 × 10^6^ cells were stimulated.Add 5 μL Cytostim, vortex, and incubate the cells for 2 h at 37 °C, 5% CO_2_.Add 245.5 μL of culture medium and 0.5 μL Brefeldin A. vortex and incubate the cells overnight (14 h) at 37 °C, 5% CO_2_.The following morning, stop the stimulation with 100 μL EDTA 20nM, vortex, and incubate for 10 min at room temperature.Add 3 mL PBS + 1% FBS (4 °C).Centrifuge the cells for 5 min at 400× *g*, then discard the supernatant and vortex gently.Continue with the intracellular staining protocol.

### 4.9. Intracellular Staining (ICS) for Flow Cytometry

Following incubation, cells were incubated for 30 min at 4 °C in the dark with the cocktail of antibodies for surface makers staining (Appendix A). Subsequently, cells were directly fixed and permeabilized with 250 μL Cytofix/Cytoperm solution and incubated for 20 min. Of note, the wash step was eliminated to avoid losing cells. The cell suspension was centrifuged for 5 min at 400× *g*, and the supernatant was discarded. Then, cells were washed with 1 mL of Perm/Wash buffer and centrifuged for 5 min at 400× *g* twice. Once the supernatant was discarded, cells were stained with intracellular antibodies (Appendix A) and incubated for 30 min at 4 °C in the dark. Finally, the cell suspension was washed with 1 mL of Perm/Wash buffer, centrifuged for 5 min at 400× *g*, and resuspended in 500 μL of Running Buffer.

### 4.10. Whole Blood Antibody Staining

Fresh whole blood (100 μL) was stained with monoclonal, fluorescent-labeled antibodies (Appendix A). Cells were incubated for 20 min at RT in the dark and lysed with FACS Lysing Buffer (BD Beckton Dickinson, Franklin Lakes, USA, New Jersey) according to the manufacturer’s instructions. After a washing step with PBS + 1%, FBS pellets were resuspended in 300 μL of Running Buffer (Miltenyi Biotec, Bergisch Gladbach, Germany) before acquisition within 1–4 h.

### 4.11. Acquisition and Data Analysis

Following cell staining, samples were acquired in a BD LSR Fortessa SORP cytometer. The FACS tubes were acquired completely. Spectral overlap compensation between all channels was done automatically by the BD FACSDiva software v8.0.1 (BD Biosciences, Franklin Lakes, USA, New Jersey) using single-color controls. Comparable day-to-day performance of the cytometer was ascertained by running CS&T calibration beads (BD) weekly. In addition, for standardization of instrument settings longitudinally, an 8-peak Rainbow Compensation Particles Set (BD) was used prior to every experiment, and photomultiplier tubes (PMTs) voltages were adjusted if needed. Flow cytometry data were analyzed using FlowJo v10.8.1 (TreeStar, Portland, USA, OR). The FlowJo boolean gating tool was used to create the co-expression profiles.

### 4.12. Statistical Analysis

GraphPad Prism (version 8.0, GraphPad Software, Inc., La Jolla, CA, USA) was used for graph representation and statistical analysis. Data are represented as individual values with the median and 25th to 75th percentiles. Each dot represented a donor/patient. For data normal distribution assessment, the Shapiro–Wilk test was used. According to this, the Kruskal–Wallis’s test (for multiple group comparison) and Mann–Whitney test (for comparison of sample pairs) were used to derive *p*-values for data comparison among groups. Significance was indicated by * *p*-value < 0.05, ** *p*-value < 0.01, *** *p*-value < 0.001, and **** *p*-value < 0.0001.

## 5. Conclusions

Our data show the benefits of using a cytometry-based protocol for the study of cardiovascular, and other tissues’, immunopathology. An in-depth characterization of the cell populations implicated in AS will allow the identification of new biomarkers of early diagnosis and the development of new therapies for the treatment of this disease.

## Figures and Tables

**Figure 1 ijms-24-02194-f001:**
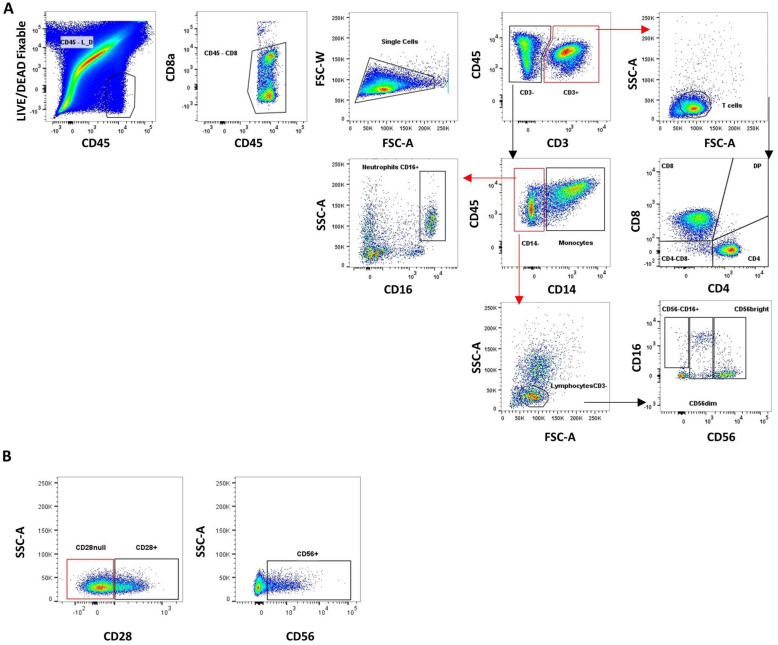
Gating strategy for aortic valve infiltrate. (**A**) After cleaning strategy, CD3+ cells and CD3− cells were gated in a CD3 vs. CD45 plot. From the CD3+ gate, any remaining debris was excluded an SSC-A vs. FSC-A plot. Then CD4+, CD8+, CD4−CD8−, and CD4+CD8+ (DP) T cell subsets were determined in a CD8 vs. CD4 plot. From the CD3− gate, monocytes (CD14+) and CD14− cells were gated in an SSC-A vs. CD14 plot. From the CD14− gate, CD16+ neutrophils were gated in an SSC-A vs. CD16 plot, and CD3− lymphocytes were gated in an SSC-A vs. FSC-A plot. Finally, from CD3− lymphocytes gate, NK cells and their subsets were defined confronting CD56 vs. CD16. (**B**) From CD4+, CD8+, and CD4−CD8− T cells, CD28 and CD56 gate were set by confronting SSC-A vs. CD28 or CD56.

**Figure 2 ijms-24-02194-f002:**
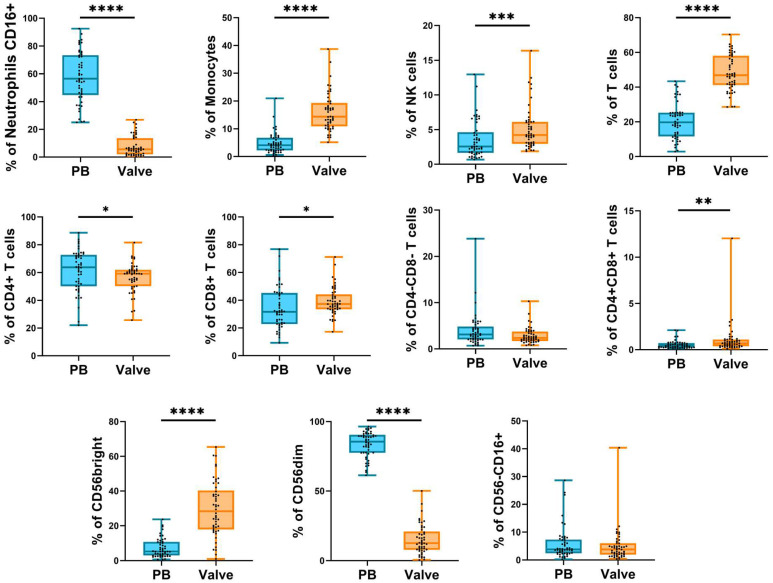
Frequencies of innate and adaptative leukocyte subpopulations. Boxplot graphs showing the frequencies of innate and adaptative cell subsets from peripheral blood (blue) and valvular infiltrate (orange). Frequencies were calculated from the total number of leukocytes. The horizontal bar shows the median and whiskers show the maximum and minimum values. Each dot represents a donor. The significance of the data was determined by the Mann–Whitney comparison test. * *p* < 0.05, ** *p* < 0.01, *** *p* < 0.001, and **** *p* < 0.0001.

**Figure 3 ijms-24-02194-f003:**
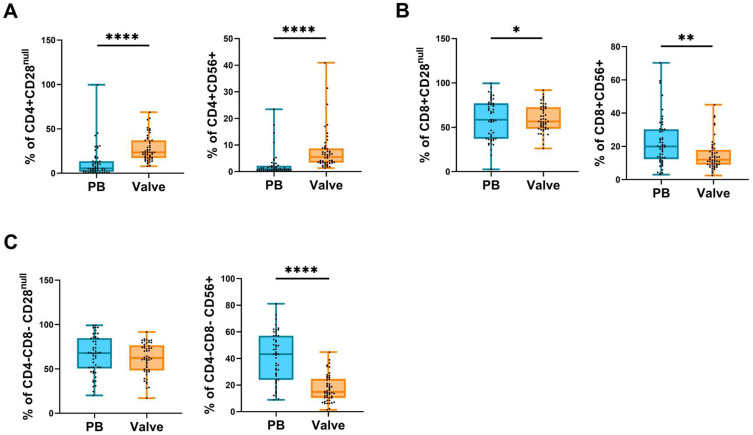
Expression of CD28 and CD56 in T cells. Boxplot graphs showing the phenotype of (**A**) CD4+, (**B**) CD8+, and (**C**) CD4−CD8− T cells from peripheral blood (blue) and valvular infiltrate (orange). The horizontal bar shows the median and whiskers show the maximum and minimum values. Each dot represents a donor. The significance of the data was determined by the Mann–Whitney comparison test. * *p* < 0.05, ** *p* < 0.01, and **** *p* <0.0001.

**Figure 4 ijms-24-02194-f004:**
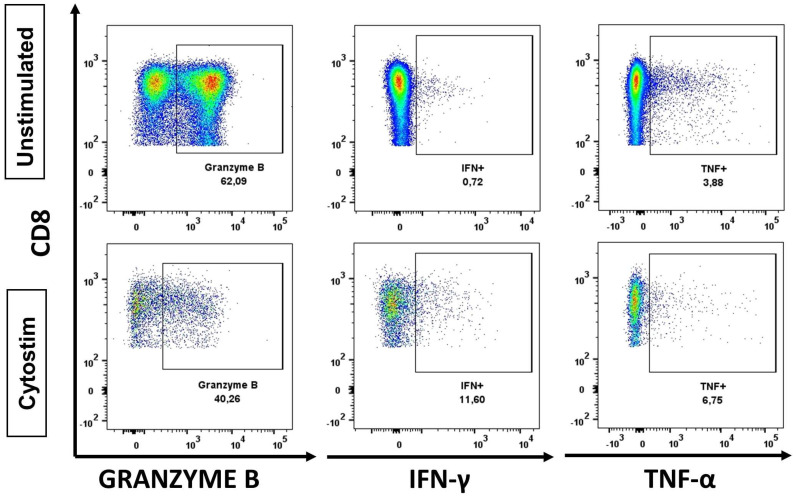
Granzyme B, IFN-γ, and TNF-α expression in CD8+ T cells. Dot plots show Granzyme B, IFN-γ, and TNF-α expression in CD8+ T cells in unstimulated PBMCs, and valvular infiltrated cells stimulated with Cytostim from the same donor.

**Table 1 ijms-24-02194-t001:** Demographics of studied individuals (*n* = 50).

Group Name	Sex (Male/Female)	Age (Mean ± SD)	*n*
Aortic stenosis	34/16	42–79 (67 ± 8.76)	50

## Data Availability

De-identified Case and Controls datasets used and/or analysed during the current study are available from the corresponding author on reasonable request.

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
