# Peer review of "A Straightforward Cytometry-Based Protocol for the Comprehensive Analysis of the Inflammatory Valve Infiltrate in Aortic Stenosis"

_ijms, 2023, doi:10.3390/ijms24032194_

Round 1
Reviewer 1 Report
Aortic valve stenosis (AVS) is an active inflammatory atherosclerotic-like process, but AVS can be divided into three stages: mild, moderate and severe AVS. Your study subjects had severe AVS (19 calcified human aortic tricuspid valves). What about inflammatory leukocyte level infiltration in aortic valve explants in mild to moderate AVS?
Author Response
We would like to thank the reviewer for their positive assessment of our work. We agree that AVS is a progressive pathology, and it would be very interesting to study the leukocyte infiltration in in mild to moderate patients. However, mild, and moderate AVS are controlled using drugs and medical follow up. Current European and international guidelines only consider the replacement of the stenotic valve (surgery) if the stage is severe. Therefore, it has been impossible to obtain the corresponding tissue.
Reviewer 2 Report
First of all, I would like to thank the authors of this paper. I found it to be quite an interesting read and I always welcome papers, which are exploring new/better techniques for patient care.
Authors of the article have developed a straightforward cytometry-based protocol for the comprehensive analysis of the inflammatory valve infiltrate in aortic stenosis. Authors have included 19 calcified human aortic tricuspid valves into the study, which is not valuable enough to prove their thesis. Work would be more valuable with larger sample size to validate the findings.
Beginning of the manuscript starts with a transparent introduction. Researchers gave us a very profound material and methods description. Authors clearly emphasized inclusion criteria for the study group, but they didn’t indicate the exclusion criteria. Study was conducted in accordance with principles of the Declaration of Helsinki and informed consent was obtained from all the members of the study group. Authors have also provided us with the information that the study was approved by the Ethics Committee of Hospital Universitario Reina Sofia of Cordoba (Spain). Authors have prepared an advanced statistical analysis with legible presentation of the results.
Unfortunately, authors prepared very short discussion and brief conclusions in a very bossy and authoritative narration. You cannot say that your data show the technical superiority of your protocol over other methodologies for the study of cardiovascular issues with the representation of only 19 samples. What’s more, limitations of the study are missing.
Final suggestion regarding the manuscript: authors have prepared only 20 publications in the reference section. Some of them are dated back to 1997/2000. It would be better to moderate and expand the reference list with publications dating up to 5-6 years ago. It would improve the literature concerning the research.
Author Response
We would like to thank the reviewer for their helpful comments. We have addressed and/or discussed all the observations and suggestions made in our detailed point-by-point reply (see below).
Comment 1:
“Authors of the article have developed a straightforward cytometry-based protocol for the comprehensive analysis of the inflammatory valve infiltrate in aortic stenosis. Authors have included 19 calcified human aortic tricuspid valves into the study, which is not valuable enough to prove their thesis. Work would be more valuable with larger sample size to validate the findings”.
Other studies using immunohistochemistry techniques included a similar number of samples as our study. As we used these other protocols (references 18 and 20) as comparison with ours, the sample size originally presented seamed adequate to us. Nevertheless, since the submission of the manuscript we have continued our analysis and we have additional data for up to 50 AS patients. Thus, we have updated the results section including these new data. We hope that the reviewer will find now that the sample size is adequate to validate our protocol.
Comment 2:
“Authors clearly emphasized inclusion criteria for the study group, but they didn’t indicate the exclusion criteria”.
Inclusion and exclusion criteria were described in 2.1 subsection of Materials and Methods. We apologise if the information provided was not sufficiently detailed. In order to clarify this point, we have rewritten this paragraph to:
AS patient’s selection was done according to the following criteria: patients with aortic stenosis (isolated or in association with others cardiac diseases) with indication for surgical replacement, regardless of the anatomical characteristics of the native valve. Exclusion criteria included: chronic infections apart from CMV (HCV, HBV, HIV, etc), inflammatory disease (oncologic diseases, rheumatoid arthritis, ankylosing spondylitis, ulcerative colitis, crohn's disease, celiac disease, systemic lupus erythematosus, multiple sclerosis, psoriasis, etc.), immunosuppressor treatment or haemodialysis. antihypertensives drugs that have calcium channel blockers as their active ingredient, such as: Lercanidipine, Felodipine, Amlodipine, Bepridil, Diltiazem, Isradipine, Nicardipine, Nifedipine, Nimodipine, Verapamil, Clevidipine and Nisoldipine. (Only for those included in the functional analyses).
Comment 3:
“Unfortunately, authors prepared very short discussion and brief conclusions in a very bossy and authoritative narration. You cannot say that your data show the technical superiority of your protocol over other methodologies for the study of cardiovascular issues with the representation of only 19 samples. What’s more, limitations of the study are missing”.
We apologise for the shortness and tone of the discussion. We have added some more discussion and we have eliminated any authoritative narration. We have also included a potential limitation statement regarding the type of samples used.
Comment 4:
Final suggestion regarding the manuscript: authors have prepared only 20 publications in the reference section. Some of them are dated back to 1997/2000. It would be better to moderate and expand the reference list with publications dating up to 5-6 years ago. It would improve the literature concerning the research.
We thank the reviewer for this suggestion. We have now included 6 new up to date references that we hope will improve the bibliography presented.
We are hoping that we have sufficiently addressed all the reviewer concerns and that our responses will satisfy the reviewer.